# The neoepitope of the complement C5b-9 Membrane Attack Complex is formed by proximity of adjacent ancillary regions of C9

Charles Bayly-Jones [1,2,8], Bill H. T. Ho [1,8], Corinna Lau[3], Eleanor W. W. Leung[1], Laura D'Andrea[1],
Christopher J. Lupton [1], Susan M. Ekkel[1], Hariprasad Venugopal[4], James C. Whisstock [1,2,5],
Tom E. Mollnes[3,6,7], Bradley A. Spicer [1,2 ✉] & Michelle A. Dunstone [1 ✉]

The Membrane Attack Complex (MAC) is responsible for forming large β-barrel channels in the membranes of pathogens, such as gram-negative bacteria. Off-target MAC assembly on endogenous tissue is associated with inflammatory diseases and cancer. Accordingly, a human C5b-9 specific antibody, aE11, has been developed that detects a neoepitope exposed in C9 when it is incorporated into the C5b-9 complex, but not present in the plasma native C9. For nearly four decades aE11 has been routinely used to study complement, MAC-related inflammation, and pathophysiology. However, the identity of C9 neoepitope remains unknown. Here, we determined the cryo-EM structure of aE11 in complex with polyC9 at 3.2 Å resolution. The aE11 binding site is formed by two separate surfaces of the oligomeric C9 periphery and is therefore a discontinuous quaternary epitope. These surfaces are contributed by portions of the adjacent TSP1, LDLRA, and MACPF domains of two neighbouring C9 protomers. By substituting key antibody interacting residues to the murine orthologue, we validated the unusual binding modality of aE11. Furthermore, aE11 can recognise a partial epitope in purified monomeric C9 in vitro, albeit weakly. Taken together, our results reveal the structural basis for MAC recognition by aE11.

[1] Biomedicine Discovery Institute, Department of Biochemistry and Molecular Biology, Monash University, Melbourne, VIC, Australia. [2] ARC Centre of Excellence in Advanced Molecular Imaging, Monash University, Melbourne, Australia. [3] Research laboratory, Nordland Hospital Trust, Bodø, Norway. [4] Ramaciotti Centre for Cryo-Electron Microscopy, Monash University, Clayton 3800 VIC, Australia. [5] EMBL Australia, Monash University, Melbourne, VIC 3800, Australia. [6] Institute of Immunology, Oslo University Hospital and University of Oslo, Oslo, Norway. [7] Centre of Molecular Inflammation Research, Norwegian University of Science and Technology, Trondheim, Norway. [8]These authors contributed equally: Charles Bayly-Jones, Bill H.T. Ho. ✉email: bradley.spicer@monash.edu; michelle.dunstone@monash.edu

The complement system is an integral component of innate immunity composed of a large set of plasma and cell surface proteins responsible for the rapid detection and elimination of foreign pathogens[1]. Complement activation is initiated via the recognition of antibodies bound to their antigen, by foreign surfaces like bacterial membranes with their numerous constituents of conserved patterns, or via spontaneous cleavage of C3[2]. Recognition leads to the formation of the central catalytic C3 convertase that propagates a proteolytic cascade of post-translational modifications, driving the opsonisation of pathogens or damaged host cells[2].

This process initiates an indiscriminate immune response leading to the terminal effector pathway of complement, with the formation of the terminal C5b-9 complement complex (TCC), which exists in two different forms. The inert soluble sC5b-9 can be measured in plasma and other body fluids as indicator of complement activation and the Membrane Attack Complex (MAC) is incorporated in a lipid membrane and can lead to lysis and cell death, or to sub-lytic activation on host cells[3,4]. Activation of C5 with formation of TCC also leads to release of the potent anaphylatoxin C5a leading to inflammation[5]. Unregulated complement activation can also harm nearby tissues[6]. Therefore, to prevent damage to normal tissues, numerous aspects of complement are tightly regulated[7]. For example, host cells are protected from the terminal complement pathway by the primary MAC inhibitor, CD59[8].

Complement activation and MAC assembly culminate in the formation of membrane pores that are thought to mediate cell lysis by direct osmotic influx or the transfer of secondary effector molecules into a wide variety of pathogenic organisms[9–11]. In addition to the elimination of pathogens, the formation of C5a and MAC contribute to numerous autoimmune and inflammatory illnesses such as systemic lupus, age-related macular degeneration, transplant rejection, as well as some conditions that involve primary deficiencies of complement inhibitors, such as CD59, causing paroxysmal nocturnal haemoglobinuria[12–15]. Furthermore, the terminal stages of complement activation contribute to some of the acute and chronic inflammatory responses that result from viral infection, for example, the association of systemic complement activation with respiratory failure in COVID-19 hospitalised patients[16–18].

The MAC forms large heterogeneous pores by the sequential binding of single copies of C5b, C6, C7, C8, and up to 18 units of C9. Under physiological conditions, some have postulated that an average of 1–3 C9 molecules is sufficient for lysis[19]. Of these, C6, C7, C8, and C9 are members of the Membrane Attack Complex/Perforin/Cholesterol Dependent Cytolysin (MACPF/CDC) superfamily, a group of cytolytic pore-forming proteins[20]. Recent structural studies of the MAC and its components have provided insight into the final assembled architecture and the structural rearrangements that enable its assembly, such as the canonical unravelling of the transmembrane β-hairpins within the MACPF/CDC domain and global shifts of its ancillary domains[21]. These studies and others indicate that large scale conformational rearrangements, such as movement of TMH1 in C9, are characteristic of MAC assembly[22–24].

To visualise and quantify the level of complement activation at various stages, different diagnostics tools have been developed. These tools are typically in the form of monoclonal antibodies, which detect neoepitopes exposed upon activation and thus are present specifically in the activation products and not in their native components[4]. One such tool, the monoclonal mouse antibody aE11, uniquely recognises a neoepitope that is present in C9 only after activation, either by the assembly of TCC or by C9 polymerisation (such as polyC9), but is not present on the native soluble C9[25]. Therefore, aE11 is used to detect complement

activation at the level of the terminal cascade in vitro, in situ, and ex vivo, either as sC5b-9 by ELISA using plasma samples, or as stable MAC pores in histology and immunofluorescence samples[26–28].

To date, aE11 is one of the most widely used commercial tools to detect human TCC in vitro, in situ, and ex vivo. While aE11 is human specific, it has also been shown to cross-react with baboon, horse, and pig C9, but does not react with mouse C9[29,30]. Despite being introduced nearly four decades ago, little is known about the structural basis of the neoepitope.

In this regard, two main possibilities may give rise to the C9 neoepitope in TCC. Either extensive conformational changes in the C9 structure induced by its oligomerisation may generate novel surface topographies that define the neoepitope, for example, by release or exposure of previously occluded regions. Alternatively, the neoepitope may be defined by a discontinuous region found in TCC but not in monomeric C9. Here, a discontinuous epitope is defined by two or more non-contiguous sequences that are spatially adjacent in the tertiary structure. Of course, these modalities are not mutually exclusive and therefore, a combination of the two may similarly define the neoepitope.

The aim of this study was to assess the structural basis for the formation of the C9 neoepitope and determine whether conformational changes due to pore formation are required. For this purpose, we employed cryo-EM to resolve the structure of a complex consisting of an aE11-Fab fragment bound to polyC9 as a mimic of MAC. Further, we use site-directed mutagenesis and surface plasmon resonance to study the impact of key residues on epitope presentation and aE11 recognition.

## Results

### Structural characterisation of the C9 neoepitope in oligomeric C9 by cryo-EM.
To investigate the molecular binding determinants of the C9 neoepitope-specific monoclonal antibody aE11, we sought to characterise the structural basis of the aE11/C9 interaction. Given that aE11 is known to recognise the oligomeric C9 component of MAC, we chose to employ the polyC9 model. PolyC9 is also recognised by aE11 and its structure closely resembles MAC[21,22,31]. For this reason, we prepared a soluble ~2.5 MDa complex (aE11-Fab/polyC9) consisting of Fab fragments of aE11 and in vitro homo-oligomerised C9 (polyC9; as a MAC mimic[22]) which was visualised by cryo-EM (Fig. 1, Supplementary Fig. 1, and Table 1).

The aE11-Fab/polyC9 complex revealed an expected symmetrical 22-subunit assembly with stoichiometric binding of aE11-Fab positioned between each C9-C9 interface (Supplementary Fig. 1). Each molecule of aE11-Fab loosely packs against two identical neighbouring molecules, anchored by the C9 neoepitope. Each aE11-Fab molecule possesses notable flexibility about the hinge region, defined by the boundary of IgG domains, observable as diffuse cryo-EM density (Supplementary Fig. 1). Further, due to the intrinsic heterogeneity of polyC9, the reconstruction was initially limited to >4 Å global resolution (Supplementary Fig. 1). To overcome this, we subsequently performed symmetry expansion and focused refinement of three C9 protomers and the variable regions of two aE11-Fabs. This increased the resolution to 3.2 Å, indicating that flexibility was substantially dampening high-resolution features. Further, the interpretability of the map was drastically improved, thereby enabling an atomic model to be built.

The neoepitope is positioned in the peripheral, upper portion of the polyC9 assembly at the interface between C9-C9 protomers as defined by the aE11-Fab/polyC9 interface (Fig. 1a–c). The interface consists of discontinuous regions contributed by the ancillary TSP and LDLRA domains from two adjacent subunits, as well as the

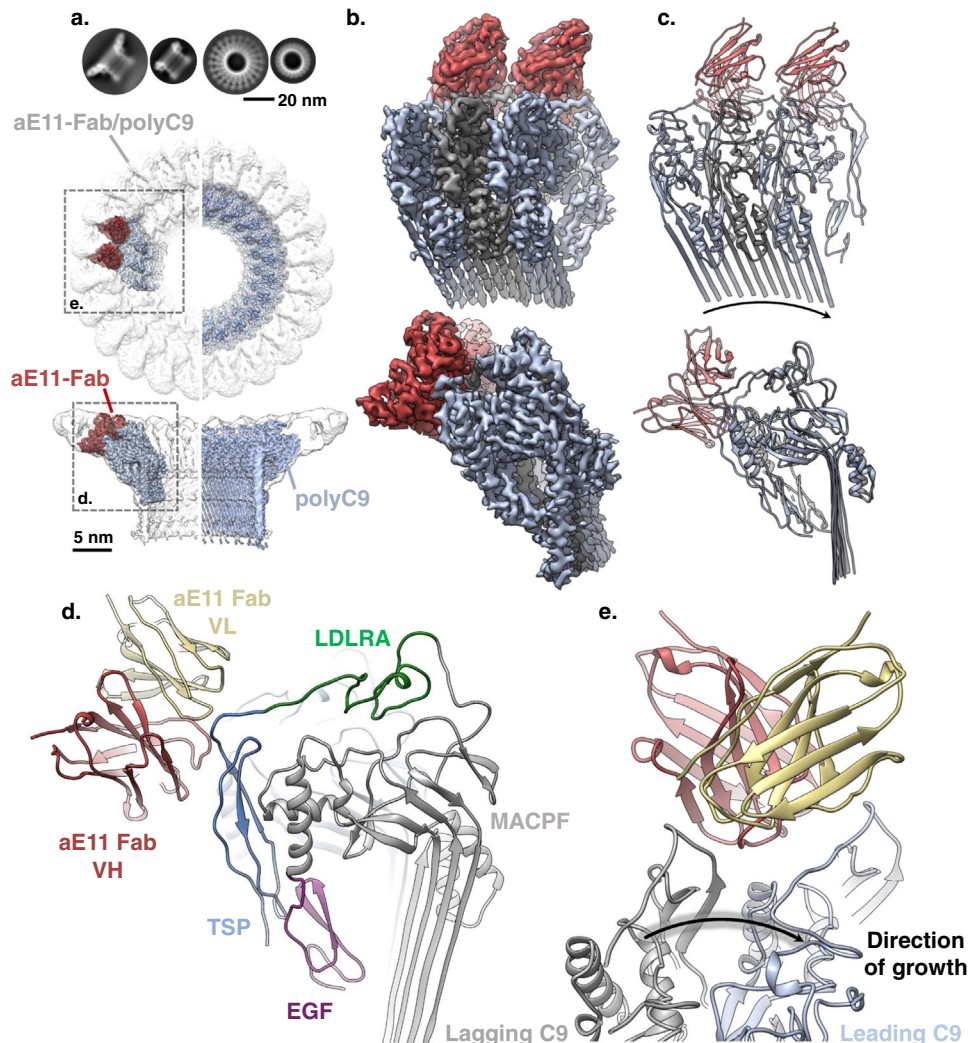

**Fig. 1 Cryo-EM reconstruction of aE11-Fab/polyC9. a** Above: comparisons of 2D class averages between polyC9 alone and aE11-Fab/polyC9. Below: 3D density map comparison between top-down and side orientations of the full cryo-EM maps of the aE11-Fab/polyC9 complex (transparent grey surface) and unbound polyC9 (blue). Left: focused refinement of C9 trimer (blue) with aE11-Fab (red) positioned in the global map of aE11-Fab/polyC9 (transparent grey). Right; unbound polyC9 alone (EMDB: 7773). **b** Focused view of the boxed region from (a; oblique and side views), three C9 subunits, and two aE11-Fabs are shown as isosurfaces. **c** Corresponding cartoon representation of the atomistic model derived from (**b**). The direction of MAC assembly is indicated by the arrow. **d** The positions of the heavy (H) and light (L) chains of aE11-Fab, in red and beige, respectively, and domains of the C9 subunit: MACPF (grey) and ancillary domains TSP (blue), LDLRA (green), and EGF (purple). **e** Top-down view of the aE11-Fab wedged between the two C9 protomers, defined as leading (light blue) and lagging (grey) subunits. Black arrow shows direction of MAC assembly.

MACPF/CDC linchpin on the leading C9 (Fig. 1d). Together these regions produce a relatively deep groove that accommodates the aE11 complementarity-determining regions (CDRs), defining a total buried surface area of 1187 Å². Assembly of the interface is, therefore dependent on the proximity of two distinct C9 chains, giving rise to a quaternary discontinuous epitope.

Comparison of the aE11-Fab/polyC9 structure with our previous polyC9 structure revealed no notable conformational differences between the bound and unbound forms at these resolutions (RMSD 0.989 Å; Supplementary Fig. 2). Notably, the aE11 bound structure possesses improved cryo-EM density for the TSP and LDLRA domains compared to our previous reconstruction (EMD-7773)[22] enabling this region to be modelled (Fig. 1d). Superposition of the murine C9 crystal structure with the aE11 bound structure shows only minor deviations in TSP and LDLRA (i.e., the region of aE11 binding), suggesting that major conformational changes between monomeric and oligomeric C9 in this region are unlikely (RMSD 1.208 Å; Supplementary Fig. 2).

Hence, neoepitope formation does not appear to be predominantly driven by conformational changes in the peripheral region of oligomeric C9. Taken together, our structure supports a quaternary discontinuous neoepitope mode of recognition.

**Structural basis of aE11 neoepitope recognition.** Together three interfaces define the quaternary discontinuous C9 neoepitope of MAC, contributed from two C9 protomers. We define these protomers by their relative position with respect to MAC assembly as either the lagging or leading C9 molecule (Fig. 1e)[21]. The leading C9 protomer possesses the largest buried surface area with aE11 (695 Å²), forming the major interface through numerous contacts with the heavy chain CDRs (Fig. 2). This surface is defined by the TSP-LDLRA loop spanning residues 68 to 82, as well as the upper portion of the MACPF/CDC linchpin helix. Conversely, the minor interface is defined by the lagging C9 protomer, contributing 492 Å² to the buried surface area, all of which originates from the opposite side of the TSP-LDLRA loop.

**Table 1 Cryo-EM data collection, refinement, and validation statistics.**

| | aE11-Fab/polyC9 (EMDB-27385) (PDB 8DE6) | aE11-Fab/ polyC9 (EMDB-28863) |
|---|---|---|
| **Data collection and processing** | | |
| Magnification | 105,000 | 105,000 |
| Voltage (kV) | 300 | 300 |
| Electron exposure (e-/Å$^2$) | 52.8 | 52.8 |
| Defocus range (μm) | −0.5 to −2.0 | −0.5 to −2.0 |
| Pixel size (Å) | 1.4 | 1.4 |
| Symmetry imposed [after expansion] | [C1] | C22 |
| Initial particle images (no.) [symmetry expand] | 48248 [1,061,456] | 48248 |
| Final particle images (no.) [symmetry expand] | 10,061 [221,346] | 7783 |
| Map resolution (Å) | 3.2 | 4.2 |
| FSC threshold | 0.143 | 0.143 |
| Map resolution range (Å) | 3.0 to 5.0 | 3.8 to 7.0 |
| **Refinement** | | |
| Initial model used (PDB code) | 6DLW/3BAE | |
| Model resolution (Å) | 3.35 | |
| FSC threshold | 0.5 | |
| Map sharpening B factor (Å$^2$) | −134.7 | |
| Model composition | | |
| Non-hydrogen atoms | 13243 | |
| Protein residues | 1659 | |
| Ligands | 12 | |
| B factors (Å$^2$) | | |
| Protein | 22.40/163.79/79.49 | |
| Ligand | 69.84/137.08/100.74 | |
| R.m.s. deviations | | |
| Bond lengths (Å) | 0.010 | |
| Bond angles (°) | 0.895 | |
| Validation | | |
| MolProbity score | 1.81 | |
| Clashscore | 4.54 | |
| Poor rotamers (%) | 2.53 | |
| Ramachandran plot | | |
| Favoured (%) | 96.02 | |
| Allowed (%) | 3.98 | |
| Disallowed (%) | 0.00 | |

This minor interface predominantly interacts with the light chain CDRs, but not exclusively.

To assign the primary structure of aE11 to the cryo-EM density, the amino acid composition of the aE11-CDRs was retrieved by antibody variable domain sequencing using the original hybridoma cell line (Fig. 2a). The aE11-CDRs are rich in both aromatic and polar residues. Overall, the aE11-CDRs produce two distinct negative and positive interfaces, which have striking and extensive charge complementary to the major and minor interfaces of the C9 neoepitope (Supplementary Fig. 3).

In addition to global charge complementarity, two surface exposed hydrophobic regions form close contacts with aE11. Together CDR H3 and L3 form a hydrophobic pocket that accommodates a surface exposed hydrophobic loop present on the lagging C9. This pocket shields V68 of C9 forming a close hydrophobic interaction with L113 of aE11 L3 (Fig. 2c, d). Similarly, L423 present on the tip of the MACPF/CDC linchpin α-helix of the leading C9, forms hydrophobic interactions with V50 of CDR H1. Moreover, numerous CH-π interactions contribute to aE11 binding, for example between Y120 of CDR H3 and L423 or between Y112 of CDR L3 and P72, from the leading and lagging C9 interfaces respectively (Fig. 2c–e).

Mostly the interface contacts are polar, including three salt bridges that mediate interactions across the discontinuity of the C9 neoepitope (Fig. 2c–e). CDR H3 is relatively long and extends across both the leading and lagging interfaces. On the leading C9 interface, CDR H3 possesses two buried arginine residues that both contribute salt bridges (R116-E71 and R118-D78) with the predominantly negative C9 TSP-LDLRA loop (Fig. 2c–e). In addition, CDR H2 D73 contributes a third salt bridge to R65 positioned on the TSP domain of the lagging C9 (Fig. 2c).

Notably, the CDR L1 loop does not largely contribute to the aE11-Fab/C9 complex. Similarly, CDR L2 only contributes minor polar contacts with the leading C9 interface. In entirety, the interface possesses 11 unique non-covalent contacts (Supplementary Table 1), of which 4 are contributed by the light chain CDRs of aE11. In total, roughly a third (13 of 36 interface residues; Supplementary Table 1) of the interface is defined by the light chain.

**Characterisation of aE11 binding to polyC9 by surface plasmon resonance.** To validate our molecular model of aE11-Fab/ polyC9 binding, we sought to characterise the binding of aE11 to several variants of C9 with specific amino acid substitutions. The recognition of C9 by aE11 is species dependent, showing cross-reactivity against human, baboons, and pig homologues, but not murine (Fig. 2f and Supplementary Fig. 4)[29,30]. Sequence alignments of the linear regions defining the neoepitope revealed poorly conserved residues between human and murine homologues, which we reasoned may underpin the molecular basis for aE11 species specificity (Fig. 2g). Based on comparison between the monomeric crystal structure of murine C9 and the structure of the human aE11-Fab/polyC9, we designed three point-mutants in the C9 neoepitope by replacing human residues with the murine counterparts, namely R65Q, V68E, and P72E. Murine residues were chosen such that the impact on the local epitope fold would be minimised, assuming that human C9 should tolerate murine C9 substitutions.

Functional characterisation of the C9 variants indicated that the point mutations retained the capacity to form MAC, with only marginal impact on haemolytic activity (EC$_{50}$) relative to the wild type (Fig. 3a and Supplementary Fig. 5). Furthermore, the mutants retained the capacity to form polyC9 as assessed by negative stain EM and possessed similar thermal stability (apparent melting temperature, T$_M$; Fig. 3b, c). These results suggest the human-to-murine mutations have little impact on the capacity to form oligomeric C9. This is consistent with the knowledge that murine C9 ultimately forms the polymeric component of the murine MAC. On the basis of these analyses alone, we cannot fully disregard the possibility that these mutations cause subtle local changes in the epitope.

We next performed a series of SPR experiments, by immobilising aE11 IgG onto a CM5 chip and introducing polyC9 as the analyte. We acknowledge two major limitations to this approach. Firstly, these estimates make the simplifying assumption of a homogeneous oligomeric 22-mer of polyC9. However, we frequently see broken or aggregated oligomeric C9 by negative stain EM (Fig. 3b). Secondly, the binding kinetics of aE11 to polyC9 do not obey true one-to-one binding. As such, a more robust estimate for affinity would require the inverse experiment, whereby polyC9 is immobilised and aE11 Fab is measured as the analyte. We were unable to quantify the affinity of the interaction between aE11 and all variants with this method as the necessary quantities of antibody Fab precluded us from performing these experiments.

Instead, for comparisons between C9 mutants, we used the maximum binding response as a relative measure of binding

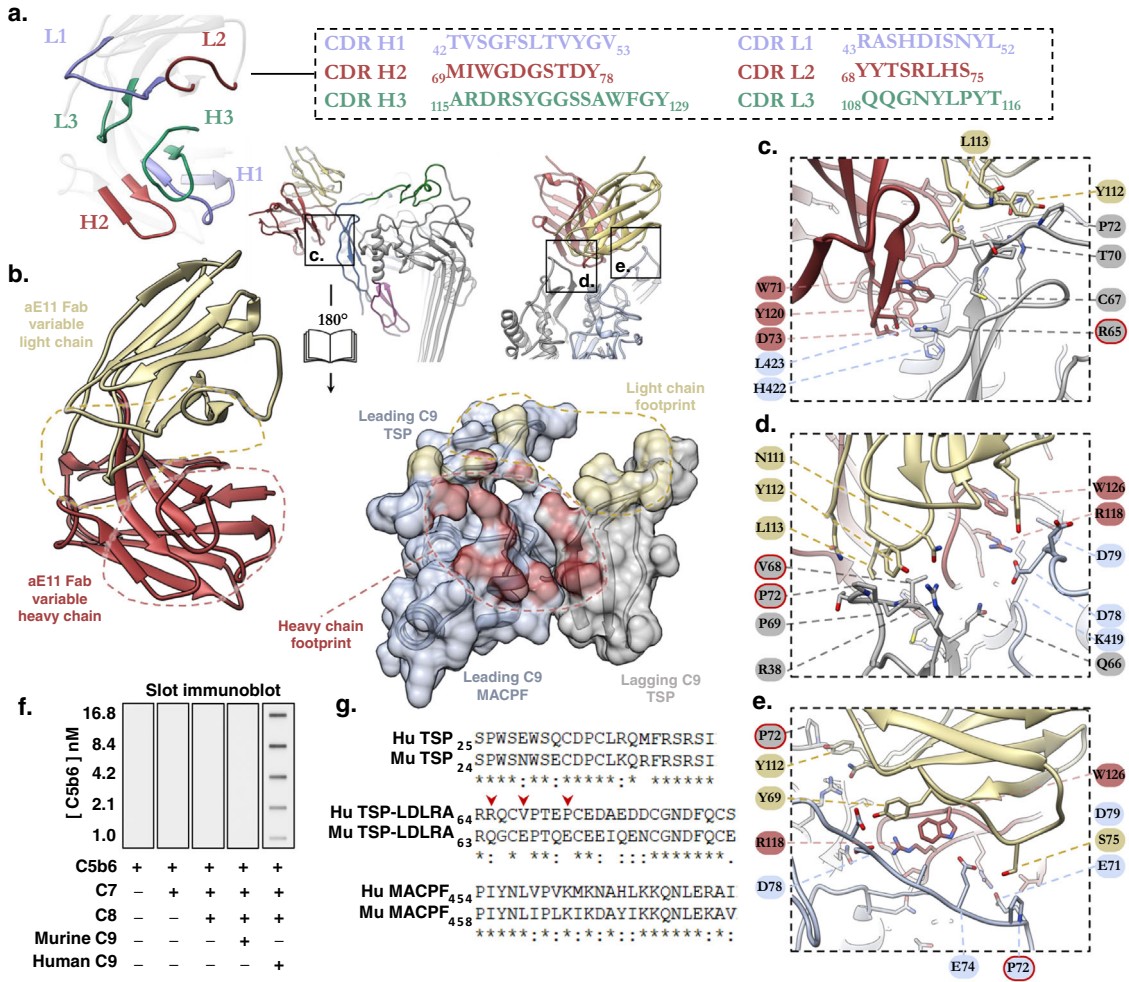

**Fig. 2 Structural basis of aE11-Fab/polyC9 binding. a** Illustration of the aE11-CDRs (coloured, blue, green, red) on the antibody and its corresponding primary sequence. **b** Focused view of the opened aE11-polyC9 interface (split by 180°) that defines the MAC neoepitope. Left: close-up structure of the aE11-CDRs. Right: Transparent surface and cartoon illustration of the leading (light blue) and lagging (grey) C9 subunits, where the footprints of heavy and light chain binding regions are shaded red and yellow, respectively. A dashed line illustrates the buried surfaces of the interface. **c–e** Key regions of antibody binding to C9. Locations of these regions correspond to the boxed regions in (**b**). Mutated residues for validation studies are encircled in a red outline. **f** Slot immunoblot of aE11 binding to the oligomeric C9 component of whole MAC. Oligomeric human C9 is recognised by aE11, but not oligomeric mouse C9. MAC assembly intermediates (C5b6, C5b-7, C5b-8) are included as controls. Concentrations of C5b6 are marked for each condition. The C7, C8αβγ, and C9 concentrations were constant at 30, 30, and 7 nM, respectively. **g** Sequence alignment of the linear regions of human and murine C9 that contribute to the quaternary discontinuous epitope.

compared to wild type C9 (Fig. 3d, e and Supplementary Fig. 6). Mutagenesis of a single salt bridge formed between C9 and aE11-CDR H2, R65Q, resulted in only attenuated binding of aE11. Conversely, V68E, which introduces a charge repulsion with the opposing D117 of aE11-CDR H3, resulted in the most substantial loss of binding. Unexpectedly, P72E alone had no impact on aE11 binding when compared to the wild-type C9. The combined mutation of R65Q/V68E/P72E completely abolished binding of aE11 to polyC9, resulting in a murine-like binding profile as anticipated. Taken together, these data are consistent with the quaternary discontinuous epitope observed in the cryo-EM structure and recapitulate the expected species specificity.

Disregarding the effects of multivalency in polyC9, we estimated the binding affinity using a one-to-one kinetic model to be on the order of single digit nanomolar concentration for aE11 and wild type oligomeric C9. We note that the off-rate of polyC9 was on the order of $10^{-6}$ s$^{-1}$, indicative of a very tight interaction. While the one-to-one model roughly fitted the data, we expect this approximation yields an overestimate, as avidity

and multivalency introduce a systematic error that leads to an inflated measurement.

**The partial C9 neoepitope is weakly recognised in purified C9 monomers in vitro.** Our structural model suggests that only small conformational transitions occur between monomeric and oligomeric C9 in the neoepitope region. As such, we reasoned aE11 binding may occur without the need for C9 to oligomerise. We initially hypothesised that in solution the two ancillary domains of C9 may come into sufficient proximity through transient C9-C9 interactions to enable aE11 binding. We, therefore, employed the disulphide-trapped C9 variant (F262C/V405C) that prevents C9 oligomerisation by locking TMH1. A slot immunoblot assay against monomeric C9 revealed apparent binding of aE11 to both wild type and disulphide-trapped variants (Fig. 4a). Overall, we observed less binding of aE11 to the disulphide locked mutant relative to the wild-type C9, suggesting the reduced capacity of this mutant to form C9/C9 interactions

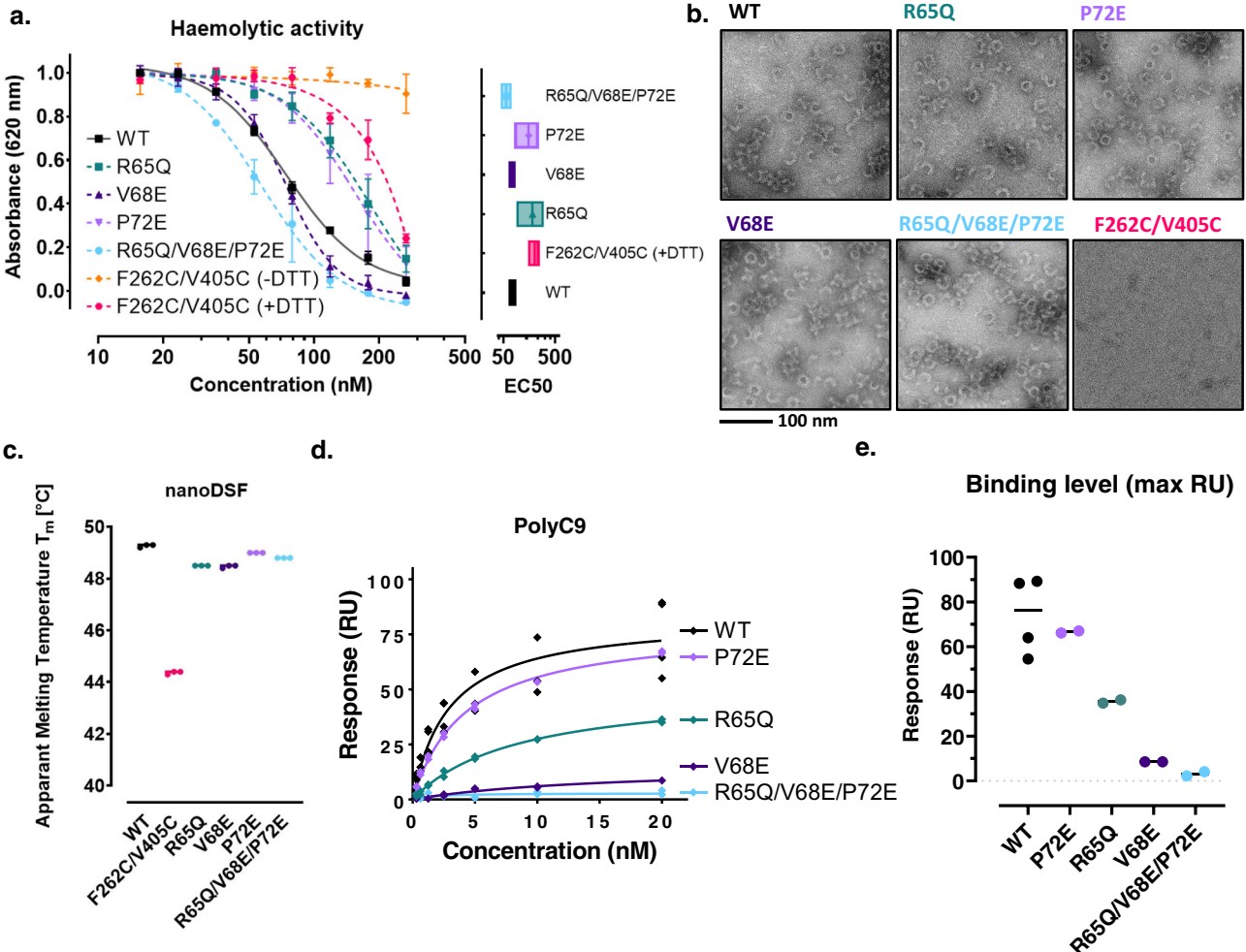

**Fig. 3 Characterisation of the C9 neoepitope by site-directed mutagenesis and SPR. a** Concentration series and effective half-maximal concentration of MAC haemolytic assay with C9 variants ($n = 3$; mean and standard error of the mean [SEM] are plotted). **b** Negative-staining electron microscopy of polyC9 variants. **c** Nano-differential scanning fluorimetry measurements of apparent thermal stability. **d** Maximal SPR response of aE11 IgG binding versus concentration of polyC9 and variants ($n = 2$). **e** Maximal binding of aE11 IgG (arbitrary response units) at 20 nM of polyC9 and variants ($n = 2$). Individual data points and their means are plotted.

may be responsible for the attenuated binding (Fig. 4a and Supplementary Fig. 7).

These findings suggest that aE11 is capable of binding purified monomeric C9 in solution, either by weakly recognising the partial neoepitope in individual C9 monomers or due to transient C9/C9 interactions in solution which assemble the full neoepitope (albeit briefly). To assess the extent, if any, of transient dimerisation, we conducted analytical ultracentrifugation (AUC) and mass photometry (MP; Fig. 4b, c). We observed no evidence of higher-order oligomers across both micro- and nanomolar concentration ranges (AUC ~7.5 μM, MP ~10 nM). Together, these findings support a model of weak recognition of a partial C9 neoepitope without transient dimerisation of C9.

Therefore, to further analyse the kinetics behind this interaction, monomeric C9 and aE11 binding was assessed using SPR (Fig. 4d–f). All C9 variants were observed to form moderate interactions with aE11, with dissociation rates on the order of $10^{-4}$ s$^{-1}$ (Fig. 4d–f and Supplementary Fig. 7). While considerable, this remains roughly two orders of magnitude weaker than binding to oligomeric C9 (Supplementary Fig. 6). Indeed, we observed a linear relationship between maximum binding and concentration, indicating steady state could not be reached despite the concentration exceeding 10-fold that at which

polyC9 saturated binding (Fig. 4d–f). These data indicate the affinity is in the low micromolar range (~$K_D$ approx. 0.5–2 μM), consistent with estimates by dot immunoblot (Fig. 4a).

We next inspected the effect of the human-to-murine residue swap mutations. Mutation of R65Q had the most profound effect on aE11 binding to monomeric C9, almost reducing binding to the level of the triple mutant (R65Q/V68E/P72E; Fig. 4d). This is followed by the P72E and V68E mutations which reduced binding two-fold. Notably, P72E did not significantly affect binding of aE11 to polyC9 indicating the partial epitope is weaker and hence more susceptible to abrogation. Consistent with earlier immunoblots, we also observed reduced binding of aE11 to the disulphide-trapped C9 (Fig. 4d–f), which had the highest apparent dissociation rate of all mutants (excluding the triple mutant; Fig. 4d–f). In the absence of transient dimerisation, it is difficult to reconcile the reduced aE11 binding to this variant. We speculate that subtle conformational differences may diminish the capacity of aE11 to bind to this variant (indeed the thermal stability of this variant is reduced; Fig. 3c).

Previous studies report that aE11 is unable to detect monomeric C9 in serum[4,25]. Thus, to reconcile this unexpected finding, we conducted a slot immunoblot assay to compare recombinant and native C9, both in vitro and in the context of

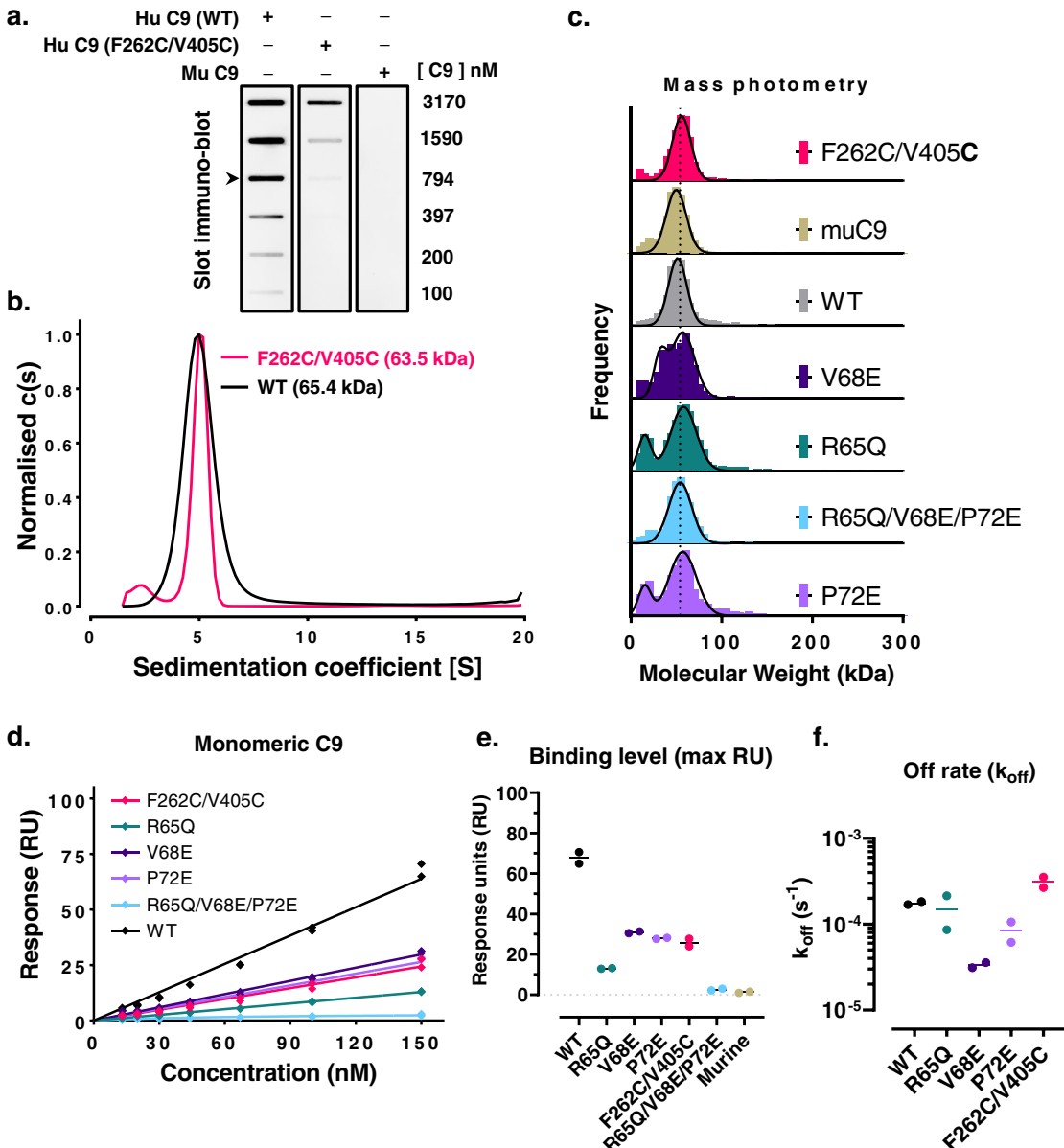

**Fig. 4 Monomeric C9 is weakly recognised by aE11. a** A concentration series of human, murine, and disulphide trapped C9 variants in the monomeric state detected by aE11 IgG in a slot immunoblot assay. An arrow shows the approximate physiological concentration of monomeric C9 ($900 \pm 200$ nM)[59,60]. **b** Analytical ultracentrifugation of wild type and disulphide-trapped C9 at micromolar concentration shows only monomeric C9 species. **c** Mass photometry measurements of the molecular mass of C9 variants corroborates AUC measurements ($n = 2$). Lower molecular weight peaks (~10 kDa) of R65Q and P72E variants correspond to a known contaminant. **d** Maximal SPR response of aE11 IgG binding versus concentration of monomeric C9 and variants ($n = 2$). **e** Maximal binding of aE11 IgG to monomeric C9 at 150 nM ($n = 2$). **f** Kinetic fit of off-rate for one-to-one binding of aE11 IgG to monomeric C9 ($n = 2$). Individual data points and their means are plotted.

human serum. As expected, aE11 recognised both recombinant and native C9 with comparable levels in vitro (Supplementary Fig. 8). Conversely, aE11 did not detect native C9 in serum, nor could it recognise recombinant C9 that had been supplemented back into C9 depleted serum at equivalent concentrations (Supplementary Fig. 8). This indicates that the weak binding of aE11 to monomeric C9 is abolished in the context of serum.

## Discussion
The MAC is a key immune effector in pathogen elimination and has been associated with a variety of different inflammatory and autoimmune diseases. As such, quantifying the level of pores

in vivo and in vitro provides insight into the contributions of the MAC to these diseases. The monoclonal antibody, aE11, recognises a neoepitope in the oligomeric C9 component of the MAC, and therefore acts as a specific marker for the quantification of MAC (simplified schematic; Fig. 5). Historically, aE11 has been used to detect C5b-9 deposition in a number of diseases, both experimentally and clinically. For example, aE11 has been used to detect MAC deposition associated with cancer, bowel, neurological, and kidney diseases[32–38], as well as MAC in macrophage inflammasome activation[39]. Finally, in the development of the total complement activity ELISA, aE11 was compared with several other anti-C9 neoepitope antibodies, and found to be the most specific and the one with highest affinity[40].

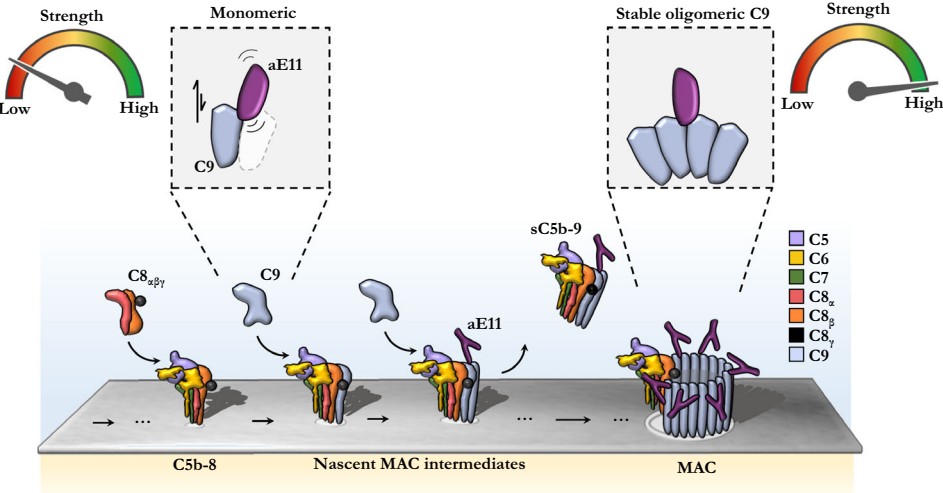

**Fig. 5 Model of aE11 binding to monomeric and oligomeric C9.** Illustration of aE11 recognition of the MAC. Under low concentration conditions little or no aE11 binding occurs. In addition, murine C9, which has a different aE11 binding interface to human C9, results in disruption of aE11 recognition. At high concentrations of monomeric C9, the partial neoepitope is weakly recognised by aE11. Stable formation of oligomeric C9 produces a long-lived, stable quaternary discontinuous epitope. Binding of aE11 to the oligomeric C9 component of the MAC is very strong.

To characterise the structural basis of aE11 recognition of the MAC, we performed a structural and biophysical analysis of the C9 neoepitope. The 3.2 Å structure of the artificial MAC mimic polyC9 in complex with aE11-Fab reveals the C9 neoepitope is formed by the proximity of adjacent C9 protomers. By comparison between the here presented aE11-Fab/polyC9 complex and polyC9 alone, we did not observe any significant structural rearrangements in the regions that contribute to the aE11-Fab/polyC9 interface that could explain the emergence of the neoepitope. We conclude the neoepitope is not defined by a conformational change but by distinct interfaces that are discontinuous in the monomeric state. These interfaces are defined by the C9 TSP-LDLRA linker region and the MACPF linchpin α-helix, which lie on the faces of the lagging and leading C9 protomers respectively. Notably, however, aE11 is also known to weakly recognise C5b-8α, due to cross-reactivity in a neoepitope on the α-chain of C8, consistent with the common domains between C9 and C8α[41].

Lastly, we unexpectedly observed aE11 is also capable of binding to purified soluble monomeric C9 in a concentration-dependent manner in vitro, albeit at much lower affinity (summarised in Fig. 5). This interaction was furthermore found to be specific as site-directed mutagenesis to the C9 neoepitope had predictable and expected impact on the interaction. While we were able to detect this interaction at physiological concentrations ([C9] in plasma ~900 ± 200 nM), we note studies, which report that plasma C9 is undetectable by aE11[4,25,40]. Indeed, we observed a serum-dependent loss of binding when supplementing high concentrations of monomeric C9 back into C9-depleted serum (Supplementary Fig. 8). This is consistent with observations in the literature[26,40,42], however, the exact cause of this effect remains to be characterised.

We validated our structural model by site-directed mutagenesis of the neoepitope and identified three critical residues in human C9 (R65, V68, and P72) that contribute to aE11 specificity for the human homologue. Mutagenesis of residues across the discontinuous interface had varying impact on aE11 binding depending on the oligomeric state of C9. Specifically, stable formation of oligomeric C9 reduced the impact of these mutations on aE11 binding, while monomeric C9 mutants were less proficient at forming a partial neoepitope. These data suggest the

combination of interfaces provides heightened aE11 binding strength indicating a level of cooperativity, in support of our quaternary discontinuous epitope model of binding.

As we have discovered that the antibody binds to the periphery of the pore, we expect that this antibody alone does not contribute to obstructing pore formation. Foundations for producing a potentially inhibitory antibody may involve binding to a major interface (e.g., the MACPF/CDC) to block oligomerisation, or capturing a conformational change. As of recent, it seems that inhibition of C7 is the last step in the terminal pathway that can inhibit lysis efficiently[43].

Our results confirm that the MAC is multivalent for the C9 neoepitope, consequently, aE11 detection in vitro and in vivo may not be a clear indication of absolute binding to the fully formed MAC. For immunofluorescence and histology studies, qualitative results may also signify the presence of C9 aggregation. To improve the detection sensitivity and accuracy of sC5b-9 levels in plasma, a sandwich ELISA has been implemented to detect more than one component of the heterogeneous complex[4,26,42]. This multiplex ELISA approach and the aE11 approach both pose a limitation for distinguishing fully assembled C5b-9 containing one or more C9 molecules from its C5b-8 precursor. Given that C5b-8 lead to a sublytic attack with roles in cellular signalling, quantitative tools that distinguish between C5b-8 and C5b-9$_n$ may provide insight into the contributions of complement to various cellular responses.

Similarly, since aE11 lacks the ability to recognise murine C9, murine models of inflammation and disease cannot be employed to study MAC dysregulation and pathophysiology. Variants of aE11 that recognise murine MAC would be beneficial and would serve to expand the repertoire of tools to study MAC-driven inflammatory diseases[44,45]. We anticipate our findings will provide a foundation to enable such rational engineering of aE11.

## Methods

**Protein production and purification**. All C9 variants were expressed using the Expi293F HEK mammalian expression system and purified using tag-less purification methods, firstly a two-step anion exchange (HiTrap DEAE [Cytiva], followed by CHT [BioRad]), and size-exclusion chromatography (Superdex 200 16/60; GE Healthcare Life Science)[46]. Likewise, oligomerisation of C9 to polyC9 was induced by incubating 1 mg mL$^{-1}$ C9 (in 10 mM HEPES, pH 7.2, 50 mM NaCl) overnight at 37 °C[22]. Human C9 mutants R65Q, V68E, P72E, and R65Q/V68E/

P72E were cloned using QuikChange PCR from the wild-type human C9, which was originally cloned in pSecTag2A (Supplementary Table 2).

For SPR analysis, monomeric C9 samples were passed through a Superdex 200 Increase 5/150 GL size-exclusion column (GE Healthcare Life Sciences) in 10 mM HEPES-NaOH, pH 7.2, 150 mM NaCl, and 1 mM EDTA to separate monomeric C9 from polyC9 and other aggregates. Fractions of 100 µL were collected, and protein-containing fractions were pooled and concentrated using the 10 kDa molecular weight cut-off concentrator (Merck Millipore). For monomeric C9, samples were extensively dialysed into the SPR running buffer (10 mM HEPES-NaOH, pH 7.2, 150 mM NaCl, 1 mM EDTA, and 0.005% P20) at 4 °C for 3 h, then overnight in fresh buffer prior to the SPR run. Serial dilutions of monomeric samples were performed in the SPR running buffer. PolyC9 samples were prepared by reacting C9 in the polyC9 buffer (10 mM HEPES, pH 7.2, 50 mM NaCl). After, polyC9 was separated from unreacted monomeric C9 in SPR running buffer by S200 5/150 GL column.

For both SPR analysis and haemolytic assays, the concentration of monomeric C9 was determined using the NanoDrop (Thermo Fisher Scientific), where the extinction co-efficient of the wild-type and mutants was taken as 8.99 $M^{-1}$ $cm^{-1}$ (pre-determined using ProtParam[47]). The extinction co-efficient for the murine C9 was taken as 8.85 $M^{-1}$ $cm^{-1}$. For polyC9 samples, the concentrations of each fraction were tested using the Bradford assay. The molecular weight of polyC9 was taken as ~1.39 MDa, whilst monomeric C9 was taken as ~63 kDa.

Monoclonal aE11 was purified from hybridomas by protein G chromatography. Briefly, monoclonal hybridoma cell culture supernatant was concentrated using Vivacell 100 PES 30K MWCO concentrators (Sartorius, Fisher Scientific) and then bound to a HiTrap Protein G HP column (GE Healthcare). After washing, bound aE11 was eluted using 0.1 M glycine-HCl, pH 2.7, and the eluate was buffer exchanged into 1× PBS pH 7.4 (Sigma Aldrich; #D8537). Generation of aE11-Fab was performed by papain digestion and purification using the Pierce Fab digestion kit following the manufacturer's protocol. The aE11-Fab sample was concentrated by centrifugal concentrator with a MWCO of 3 kDa (Merck Millipore) and further purified by size exclusion chromatography into 20 mM Tris, pH 7.2, 150 mM NaCl on a Superdex 75 10/300 column (GE Healthcare). Fractions were assessed for purity by 15% (w/v) SDS-PAGE, and those containing pure aE11-Fab were pooled and concentrated for cryoEM studies.

**Sequence determination of aE11-CDRs.** A batch of aE11 expressing hybridoma cells were sent to Genscript (GenScript Biotech [Netherlands] B.V.) for commercial Antibody Variable Domain Sequencing following their standard operating procedures, including lysis by TRIzol® Reagent (Ambion), PrimeScriptTM 1st Strand cDNA Synthesis (Takara), sequence retrieval by RACE, cloning and sequencing. A final report was provided containing the consensus variable gene sequences with corresponding V-gene and allele retrieved by IMGT-assisted analysis of V(D)J junctions.

**Mass photometry.** Mass photometry was conducted on a TwoMP instrument (Refeyn) operating at room temperature on an active anti-vibration platform. Briefly, calibration standards were measured on the day of the experiment in 20 mM HEPES, pH 7.2, 150 mM NaCl, followed by purified recombinant variants of C9. Stock solutions of roughly 100 nM were directly diluted to 5–10 nM and measurements were repeated for a minimum of two replicates. Mass photometry images were acquired and analysed using the Refeyn Aquire$^{MP}$ and Discover$^{MP}$ packages (v2.5) respectively. Gaussian fit was performed in GraphPad Prism.

**Nano-DSF.** Nano-DSF measurements of thermal stability were conducted on a Prometheus NT.48 (NanoTemper) instrument at a protein concentration of 0.5 mg $ml^{-1}$ in 10 mM HEPES, pH 7.2, 150 mM NaCl, 1 mM EDTA. Thermal protein unfolding was monitored in increments of 1 °C $min^{-1}$ from 20 to 95 °C. The apparent melting temperature was determined as the inflection point of the ratio of absorbances at 350 and 330 nm, calculated by taking the maximum of the first derivative.

**Analytical ultracentrifugation.** Sedimentation velocity analytical ultracentrifugation experiments were performed with either wild-type or disulphide-trapped C9 at 0.5 mg $ml^{-1}$ (~7.9 µM) in 10 mM HEPES pH 7.2, 150 mM NaCl, 1 mM EDTA on a Beckman Coulter Optima analytical ultracentrifuge with An60-Ti rotor at 25,000 rpm (50,310 × $g$) at 25 °C. Buffer density, buffer viscosity, and sample partial specific volumes were calculated based on their composition in SEDNTERP[48]. Solution absorbance was collected at 280 nm, and all data and frictional ratio calculations were analysed in SEDFIT[49].

**Cryo-EM sample preparation and data collection.** A complex of aE11-Fab/polyC9 was prepared for vitrification by mixing stoichiometric quantities of polyC9 and aE11-Fab. Optimal ratios of aE11-Fab to polyC9 were determined empirically by negative-stain transmission electron microscopy (TEM). Initial grid freezing conditions and negative-stain TEM were performed on a Tecnai T12 electron microscope (Thermo Fisher Scientific). Plunge freezing was performed in liquid ethane using the Vitrobot IV System (Thermo Fisher Scientific). Briefly, Quantifoil Cu R 2/2 grids were glow-discharged for 30 s in an PELCO easiGlowTM Glow Discharge Cleaning System

(PLANO). A total of 4.5 µL of aE11-Fab/polyC9 (1 mg $ml^{-1}$ in 10 mM HEPES, 50 mM NaCl, pH 7.2) was applied to the glow discharged surface and blotted (blot time of 2.5 s, blot force of −3 and drain time of 1 s) at 4 °C and 100% relative humidity. Samples were stored under liquid nitrogen until data collection. Dose fractionated movies were collected on a Titan Krios (Thermo Fisher Scientific), equipped with a Quantum energy filter (Gatan) and Summit K2 (Gatan) direct electron detector. Data acquisition was performed in EPU (Thermo Fisher Scientific).

**Cryo-EM data analysis.** Dose fractionated movies were compressed to LZW TIFF with IMOD[50]. Beam induced motion was corrected by MotionCor2 with dose-dependent weighting to compensate for radiation damage[51]. CTF estimation was performed by CTFFIND (4.1.13)[52]. Particle picking was performed by the Laplacian of Gaussian method as implemented in RELION-3.2[53,54]. Particles were extracted in a box of 450 pixels. Initial rounds of 2D classification in cryoSPARC[55] were used to discard false positive and deformed particles. A consensus refinement was performed in RELION with C22 symmetry, which was ultimately resolution limited to 3.9 Å due to heterogeneity and flexibility (C1 refinements confirmed the point-group symmetry). Sub-particles were signal subtracted and re-boxed in RELION from symmetry expanded particles. A trimer of C9 with two aE11-Fab models was initially extracted. Non-uniformed refinement and local refinements were performed in cryoSPARC[56]. Subsequently, signal subtraction was conducted to remove diffuse signal of the β-barrel and the constant region of the aE11-Fab, which was observed to undergo conformational flexibility about the Fab hinge region. This final localised reconstruction was refined and classified in cryoSPARC to 3.34 Å. Particle polishing in RELION and CTF refinement[57] in cryoSPARC further improved the reconstruction to a final resolution of 3.17 Å. Conversion between RELION and cryoSPARC were performed with *pyem*[58]. Local resolution was calculated by windowed FSC in RELION.

**Atomic modelling.** A homology model of aE11 was generated using the primary sequence and a template Fab structure of the same isotype (PDB 3BAE, IgG 2aκ) using SWISS-MODEL. This model and a trimer of C9 from our previous reconstruction (PDB 6DLW) were rigid-body fit into the cryoEM density in UCSF chimera (v1.14). Multiple rounds of flexible refinement in ChimeraX (v1.3) and ISOLDE (v1.0) were performed, iterating between Coot and ChimeraX/Isolde to build missing regions and resolve model errors and clashes. Finally, several rounds of real space refinement and manual refinements were performed in Phenix and Coot. Model validation was performed in Phenix, MolProbity and through the wwPDB OneDep validation server.

**Surface plasmon resonance.** All SPR runs were performed at 25 °C using the Biacore T200 instrument (Cytiva Lifesciences). Full aE11 IgG, that was stored in 1× PBS, was diluted to 20–100 µg $mL^{-1}$ in 10 mM sodium acetate, pH 5.5 prior to immobilisation. The antibody was immobilised onto a Series S CM5 chip at 10 µL $min^{-1}$. To activate the surface, equimolar ratios of 0.1 M N-hydroxysuccinimide (NHS) and 0.4 M *N*-ethyl-*N*′-(3-(dimethylamino)propyl)carbodiimide (EDC) were first injected for 420 s. Antibody was then injected to allow for the reactive NHS ester to directly amine couple aE11 IgG to the activated surface. Thereafter, injection of 1 M ethanolamine for 300 s at 10 µL $min^{-1}$ was performed to block the surface, as well as remove any unbound aE11. Similar amine coupling procedure was applied to the reference surface except that no antibody was coupled. Immobilisation levels for monomeric and polyC9 runs were empirically optimised to rough values of ~7000 and ~700 RU, respectively.

All experiments were performed using the multi-cycle kinetics method. In brief, each concentration of C9 flowed at 40 µL $min^{-1}$ for 250 s of contact time, followed by a 1000 s dissociation time. After, the surface was regenerated with 25 mM NaOH mixed in running buffer at a 1:3 molar ratio, for 30 s at 30 µL $min^{-1}$.

Data from the experiments were analysed using the BIAevaluation software. For monomeric C9 experiments, a simple exponential decay was assumed (corresponding to a 1:1 binding model) to extract the off rate ($K_{off}$). Steady-state affinity plots were generated by taking the maximum response units for each variant (at the same time point) for each concentration. Raw sensorgrams for both monomeric and polyC9 runs were blank and drift corrected, i.e., double referenced.

**Slot immunoblot.** Nitrocellulose paper was pre-soaked in Tris-buffered saline (20 mM Tris-HCl, 150 mM NaCl, pH 8; 1×TBS) prior to blotting of the sample. A total of 2 µL of sample at various concentrations was added onto the nitrocellulose using the slot blot apparatus. After, the nitrocellulose was blocked with 5% skim milk powder in 1×TBS before blotting with aE11 (diluted 1:10,000 in 5% skim milk powder in 1×TBS). An additional blocking step was performed with 5% skim milk + TBS before blotting with anti-mouse HRP (1:10,000 in 5% skim milk + TBS) before a final wash with 0.05% polysorbate 20 in 1×TBS. Equal volumes of chemiluminescent reagents A and B (GE Lifesciences) was added to the nitrocellulose before exposure using the BioRad chemidoc.

**Formation of MAC on liposomes.** Liposomes of *E. coli* total lipid extract (ETL, Avanti Polar Lipids) were generated as follows. Chloroform-solubilised lipid was dried under Argon gas in a clean glass tube and then desiccated under vacuum overnight at room temperature. Subsequently, the lipid film was rehydrated and

resuspended in HBS (10 mM HEPES•NaOH, pH 7.2, 150 mM NaCl) to a final concentration of 0.5 mg mL$^{-1}$ and sonicated until large chunks were no longer visible. Finally, the lipid mixture was extruded via a polycarbonate membrane with pore size of 0.4 μm (Avanti Polar Lipid) to form unilamellar liposomes. To form MAC on liposomes, a total of 200 μL of liposome mixture (diluted 1:200 in HBS) was incubated sequentially with each MAC component followed by a 5 min incubation at 37 °C. Initially, liposomes were incubated with a C5b6 (at various final concentrations from 1 to 16.8 nM), followed by C7, C8, and lastly C9 added to a final concentration of 30, 30, and 7 nM, respectively. C5b6, C7, C8 were purchased (Complement Tech) and C9 was purified as described above. After each incubation a 10 μL sample was taken for slot immunoblot.

**Haemolytic assay**. Haemolytic assays were performed as previously described[22]. In brief, following the final wash after sensitisation of sheep erythrocytes (shE), cells were diluted to a final cell concentration of $1.87 \times 10^7$ cells mL$^{-1}$. Each reaction contained 200 μL of sensitised shEs, 5 μL of diluted C9-depleted serum (Complement Tech; diluted 1 in 5 with 1×DGHB; 2.5% (w/v) D-glucose, 5 mM HEPES, 0.15 mM Ca$^{2+}$, 0.5 mM Mg$^{2+}$, pH 7.4) and 2 μL of C9 at various concentrations (diluted in 10 mM HEPES, pH 7.2, 150 mM NaCl). C9 depleted serum and C9 was added to the individual wells in the 96-well plate before adding the shEs to initiate the reaction. The absorbance of shEs was recorded at 620 nm for every minute over 60 min. Absorbance values of each reaction were extracted at 30 min. These were plotted against concentration and a sigmoidal curve was fit with non-linear regression to extract the EC$_{50}$. Unlocking experiments were performed to confirm the activity of disulphide-trapped C9. Identical volumes of reactants were used as described above. Red blood cells were incubated with C8-depleted serum for 15 min at 37 °C, prior to the addition of C9 and a final concentration of 1 mM DTT.

**Statistics and reproducibility**. Statistical tests were performed in GraphPad Prism (v9). All-versus-all comparisons between measurements of C9 variants (SPR maximum binding, kinetic off-rates, nanoDSF, and haemolytic activity assays) were conducted by unpaired t-test. All assays (except SPR) consisted of three or more independent reactions, to account for experimental variation where possible. Each reaction was measured multiple times and averaged to account for instrument noise, these were considered a single data point. SPR measurements were conducted for minimum duplicate measurements, as these were sample limited. Over the course of the study, replicate measurements were taken using with different purified protein samples to account for batch-to-batch variation. All attempts to reproduce results gave similar outcomes.

**Reporting summary**. Further information on research design is available in the Nature Portfolio Reporting Summary linked to this article.

## Data availability

All models and maps are made available. Cryo-EM maps are deposited in the Electron Microscopy Data Bank (EMDB) under accession codes EMD-27385. Coordinates of atomic models are available from the RCSB Protein Data Bank (PDB) under accession code PDB-8DE6. All source data for Figs. 3a, c–e, 4b–f, Supplementary Figs. 5a,b, 6, and 7 are provided in Supplementary Data 1 file. Likewise, uncropped blots and gels for Figs. 2f, and 4a, Supplementary Fig. 8 are provided as Supplementary Fig. 9.

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

## Acknowledgements

CBJ and BHTH acknowledge the support of the Australian Government by way of an RTP scholarship. We acknowledge the support of the Monash Ramaciotti Centre for Electron Microscopy, the the MASSIVE supercomputer team, the Monash Proteomics and Metabolomics Facility, and the Monash Macromolecular Crystallisation Facility. In particular, we acknowledge Robert Goode for support and contributions by way of proteomic experimentation and analysis. This work was partly funded by the Australian Research Council through DP180100040 and FT150100049 research grants.

## Author contributions

C.B.J., B.H.T.H., and B.A.S. co-wrote the initial draft (writing – original draft), all authors critically reviewed and edited the final version (writing – review and editing). The research goals and aims were developed by B.A.S. and M.A.D. (conceptualisation), while experimental design and methodology were developed by C.B.J., B.T.H.H., and B.A.S. (methodology). Data analysis was performed by C.B.J., B.H.T.H., C.L., E.W.W.L., L.d.A., C.J.L., and B.A.S. (formal analysis). C.B.J., B.H.T.H., C.L., S.E., E.L., L.d.A., C.J.L., H.V., and B.A.S. conducted the research and performed the investigation. S.P.R. was specifically conducted by B.H.T.H. and E.W.W.L. Mass photometry was performed and analysed by B.H.T.H. and C.B.J. Analytical ultracentrifugation and analysis were performed by B.H.T.H., L.d.A., and C.B.J. Cryo-EM sample preparation, handling, data collection, data analysis, and model building was performed by C.B.J., C.J.L., H.V., and B.A.S. B.H.T.H., C.L., S.M.E., B.A.S., and C.B.J. performed protein production and purification. B.H.T.H. conducted haemolytic, immunoblot, and functional assays. C.L. performed aE11 mRNA isolation and CDR loop sequencing. Materials and resources were provided by J.C.W., C.L., T.E.M., and M.A.D. Data curation and maintenance were performed by C.B.J., H.V., and C.J.L. All figures were produced by C.B.J. and B.H.T.H. (visualisation). The project was co-supervised by C.B.J., B.A.S., and M.A.D. (supervision). B.A.S. and M.A.D. performed project administration, coordinating research activity. J.C.W. and M.A.D. acquired research financial support (funding).

## Competing interests

The authors declare no competing interests.
