## [Peer Review File · Communications Biology]

Reviewers' comments:

Reviewer #1 (Remarks to the Author):

Bayley-Jones et al. have solved the structural basis of the C5b-9 specific antibody, aE11, to recognize a neoepitope that evolves when complement begins to assemble its polymeric pore structure. This antibody has been used for many decades to quantify/discriminate soluble C9 and MAC incorporated C9. The authors have solved the crystal structure of the aE11-C9 polymeric complex at 3.2Å. They have shown, quite nicely, the domains and some of the residues in the polymeric C9 that comprise the neoepitope. They have also shown that the antibody does not recognize soluble monomeric C9 with the same high affinity observed when the neoepitope is formed by the interaction of two C9 monomers. They also performed a variety of analysis that demonstrated that the antibody has a high affinity/avidity for the polymeric C9 versus monomeric C9. To summarize, the authors have presented the co-crystal structure of the aE11 antibody with polymeric C9 to show how it specifically recognizes polymeric versus monomeric that results from the formation of a neoepitope formed by the interaction of two C9 molecules in the MAC. There is little to criticize about the work other than it is not entirely apparent how these studies will be employed by or will help workers that utilize this well-established antibody assay.

1. Line 332: do they mean Fig. 4 since that is the figure where binding was determined?

Reviewer #2 (Remarks to the Author):

The manuscript by Bayly-Jones describes the molecular interactions between polymeric C9 and the anti sMAC monoclonal antibody aE11. The manuscript is well written, the studies carefully performed and the results carefully analyzed.

Reviewer #3 (Remarks to the Author):

In the present manuscript the authors describe the structural basis for mAb aE11 binding to the MAC (modelled by polyC9 complexes). The novel neo-epitope of mAb aE11 is mapped using cryo-EM - it is discontinuous across two C9 protomers (TSP-LDLRA linker / MACPF linchpin α -helix, respectively) and binding of aE11 does not appear to induce or involve conformational changes. These findings were further validated using relevant point mutants of C9 in a series of functional hemolysis/SPR experiments. Lastly, the authors also demonstrated binding of aE11 to monomeric C9, albeit magnitudes weaker, as expected.

Overall, this is a very well conducted study reporting out important findings. The data yielded from the technically challenging experiments is compelling. aE11 is a unique tool in the field of complement research and likely diagnostics applications and has long been used very widely. While it is known that aE11 predominantly binds a neoepitope of MAC, this is the first description of its binding modality and weak binding of monomeric C9, which will both be of great interest to anyone who applies this mAb in their research as the findings raise valid caveats to application of this mAb in quantification of total/functional MAC.

The manuscript is very well written. The experimental detail provided is adequate for production of work. The supplementary data is also complementary for the data in the main manuscript.

A few suggestions/questions are listed below:

- 1) Are there any known or potential limitations to using polyC9 as a MAC mimic? Similarly, are there caveats to characterizing soluble complexes as opposed to membrane/lipid bilayer associated complexes? If so, it would helpful for the authors to describe these.
- 2) Figure 2 is very crowded and would benefit from additional spacing between cartoons.
- 3) It would be helpful for the general audience to give a sentence or two explanation on how to

read/interpret the C9 blot and to improve labelling of Figure 4A. It is not immediately clear that this is a slot blot and therefore difficult to interpret at first. For example, reference to observations at the different concentrations applied would be helpful.

4) Figure 4D and E – if I understand correctly, the data plotted in E is the max response (150nM) plotted in D. Small suggestion, it would be helpful if colors for each mutant were consistent between D and E as at first sight it appears the data do not correspond.

5) The discussion would benefit from citing a few recent studies that have used aE11 to detect MAC in functional experiments (e.g. detection of MAC in AChR loss in vitro / bacteria / macrophages or endothelial cells). This would add weight to the importance and impact of the work presented.

Reviewer #4 (Remarks to the Author):

The manuscript entitled “The neopeptide of the complement C5b-9 Membrane Attack Complex is formed by proximity of adjacent ancillary regions of C9”, by Bayly-Jones et al., describes the interaction between the poly-C9 structure, as a mimic of the complement MAC complex, and the monoclonal antibody aE11. The 3.2 Å resolution cryoEM structure of the complex between the complement protein C9 and the aE11 Fab is shown and used to refine a molecular model that allow the authors to identify the structural determinants of the aE11-C9 interaction and shed light in the antibody recognition mechanism. Furthermore, the interaction is thoroughly studied through biophysical methods and the use of several C9 mutant variants. Specificity of the aE11 antibody for the oligomerized C9 versus the monomeric stage is also discussed.

General comments

The manuscript shows a detailed analysis of the polyC9-aE11 complex, although it does not emphasize the scope of their conclusions. A more extensive discussion about the application of the recognition mechanism discovered could include some ideas such as engineered therapeutic antibodies to inhibit the polyC9 (MAC) formation or antibodies with enhanced affinity and/or specificity for C9 oligomers, etc.

Due to the limited applicability of this monoclonal antibody (as explained along the manuscript), the rationale for the selection of this specific antibody in this work remains unclear to me. Are there any other monoclonal C9 antibodies available with other relevant properties?

It would be helpful if the authors could discuss in more detail to what extent the present work is applicable to the physiological MAC complex. The major use of the aE11 antibody is the specific recognition of the MAC in plasma as an indicator of inflammatory and auto-immune diseases. However, the analysis is performed only with the polyC9 complex and no experiments refer to the native MAC or sC5b-9.

Specific comments

1. Line 121. I think that the sentence “The aim of this study is ... investigate the necessary conformational changes that occurs during pore assembly” is confusing. The analysis of this conformational changes is focused on the aE11 interaction surface and not on the conformational changes undergone by C9 during the pore assembly.

2. Line 127. Although the title is “Structural characterization of the C9 neopeptide in MAC by cryo-EM”, no MAC structures are shown. I think the title must refer to the actual specimen (poly-C9) instead.

3. The initial consensus refinements (C1 and C22 reconstructions of the complete poly-C9-aE11 complex) mentioned in the text could be included in the EMDB deposition as accessory maps. That would help to understand statements of the manuscript such as the "stoichiometric binding of aE11-Fab" (line 134) or the "intrinsic heterogeneity of polyC9" (line 137).

4. Figure 1a. Showing complete 2D average images (unmodified boxes used along the 2D alignment and classification) instead of cropped images, as well as the corresponding scale bars, would help to compare both structures. Diffuse densities described in lines 135-137 might be indicated in the corresponding panel to help the structure interpretation.

5. Line 165. I would recommend to start with the description of the 2D average images in the figure legend to help figure interpretation.

6. Figure 2b legend does not help to follow the panel order and description. There is no relative movement information regarding the aE11-polyC9 atomic models. Some of the colors used in panels c, d and e are not easy to distinguish. To highlight the residues selected for the C9 point mutations would ease their identification.

7. Line 227. Although the analysis of the thermal stability of the C9 mutants points to no structural alterations in the overall C9 structure, it cannot be ruled out that these mutations affect in some way the structure of the interaction interface. Some extra discussion about the predicted structural alteration due to the nature of the introduced mutations might be interesting.

8. Figures 3d and 3e shows a significantly higher SEM for the wildtype than the mutants that might be worth mentioning in the text. In, figure 3e, the dashed rectangle showing "no binding" might be misleading with respect to the expected response.

9. Figure 4b does not included the samples color code (neither in the corresponding figure legend).

10. Further comparison between the recombinant monomeric C9 and the native plasma C9 that could be responsible for the differential binding specificities to the aE11 antibody would be interesting. In this regard, comment on the relative concentrations of the recombinant C9 used in these experiments versus the native C9 concentration in plasma would help to understand the discrepancy.

11. Figure 5. It is not clear what the "Moderate" and "Strongest" panels add to the figure. Some colors (C8γ and C9) are undistinguishable. Image quality should be improved.

12. Line 360-361. Are the C9 in vitro concentrations used to see the aE11-monomeric C9 binding similar to the physiological monomeric C9 concentrations in plasma?

13. In a similar way to the analysis carried out on how the interaction of this antibody does not impair the poly-C9 (MAC) assembly, I think it would worth analyzing whether it might be helping to stabilize the C9 oligomers and, more interestingly, the MAC assembly and activation.

14. Line 365. Although the reference to the C9 protein expression and purification is included, I would at least mention the expression system used in the main text, due to its relevance.

15. Line 381. Extinction co-efficient units should be included.

16. Line 382. "Murine" should be in lowercase.

17. Line 444. Radiation damage cannot be corrected with MotionCor2.

18. Line 446. Although the pixel size is shown in one Supp Table, images box size expressed in Å help interpretation.

19. Supp Figure 1d shows a significant preferential orientation. Albeit there are frontal views in the consensus refinement (as shown in the average images in panel b), they do not seem to

contribute in the density subtracted reconstruction. Comment on this would help to understand the refinement procedure.

20. Supp Figure 3 displays an outline. In Supp Fig 3b, color code for the coulombic surface is omitted. Charge complementary discussed would be clearer if the complementary areas are indicated as they have been included in panel c.

21. Line 566. "versus" should be in italics.

22. Supp Figure 6. Colors of the lines in the legend are not easy to distinguish. Thicker lines might help to clarify.

23. Table caption in Supplementary material would help to understand abbreviations used in them.

Reviewers' comments:

Reviewer #1 (Remarks to the Author):

Bayley-Jones et al. have solved the structural basis of the C5b-9 specific antibody, aE11, to recognize a neoepitope that evolves when complement begins to assemble its polymeric pore structure. This antibody has been used for many decades to quantify/discriminate soluble C9 and MAC incorporated C9. The authors have solved the crystal structure of the aE11-C9 polymeric complex at 3.2Å. They have shown, quite nicely, the domains and some of the residues in the polymeric C9 that comprise the neoepitope. They have also shown that the antibody does not recognize soluble monomeric C9 with the same high affinity observed when the neoepitope is formed by the interaction of two C9 monomers. They also performed a variety of analysis that demonstrated that the antibody has a high affinity/avidity for the polymeric C9 versus monomeric C9. To summarize, the authors have presented the co-crystal structure of the aE11 antibody with polymeric C9 to show how it specifically recognizes polymeric versus monomeric that results from the formation of a neoepitope formed by the interaction of two C9 molecules in the MAC. There is little to criticize about the work other than it is not entirely apparent how these studies will be employed by or will help workers that utilize this well-established antibody assay.

We have added additional discussion outlining the utility of our findings, as per requests by reviewers #3 and #4.

1. Line 332: do they mean Fig. 4 since that is the figure where binding was determined?

This paragraph (and figure 5) is intended as a summary of all findings. We have added the following point for clarity and reference figure 5 in the first discussion paragraph to aid the reader in understanding the MAC pathway.

“(simplified schematic; Fig 5).”

“[...] (summarised in Fig 5).”

Reviewer #2 (Remarks to the Author):

The manuscript by Bayly-Jones describes the molecular interactions between polymeric C9 and the anti sMAC monoclonal antibody aE11. The manuscript is well written, the studies carefully performed and the results carefully analyzed.

We thank the reviewer for their time.

Reviewer #3 (Remarks to the Author):

In the present manuscript the authors describe the structural basis for mAb aE11 binding to the MAC (modelled by polyC9 complexes). The novel neo-epitope of mAb aE11 is mapped using cryo-EM - it is discontinuous across two C9 protomers (TSP-LDLRA linker / MACPF linchpin α -helix, respectively) and binding of aE11 does not appear to induce or involve conformational changes. These findings were further validated using relevant point mutants of C9 in a series of functional hemolysis/SPR experiments. Lastly, the authors also demonstrated binding of aE11 to monomeric C9, albeit magnitudes weaker, as expected.

Overall, this is a very well conducted study reporting out important findings. The data yielded from the technically challenging experiments is compelling. aE11 is a unique tool in the field of complement research and likely diagnostics applications and has long been used very widely. While it is known that aE11 predominantly binds a neoepitope of MAC, this is the first description of its binding modality and weak binding of monomeric C9, which will both be of great interest to anyone who applies this mAb in their research as the findings raise valid caveats to application of this mAb in quantification of total/functional MAC.

The manuscript is very well written. The experimental detail provided is adequate for production of work. The supplementary data is also complementary for the data in the main manuscript.

A few suggestions/questions are listed below:

1) Are there any known or potential limitations to using polyC9 as a MAC mimic?

There are notable limitations to using polyC9 (e.g. lacks interactions with the membrane), however with respect to aE11 binding polyC9 is an excellent model. The C9-C9 interface of polyC9 is an accurate representation of the C9-C9 interface in MAC (as evidenced by numerous structures c.f. PDB-6DLW and PDB-6H04).

Similarly, are there caveats to characterizing soluble complexes as opposed to membrane/lipid bilayer associated complexes?

Yes, with respect to membrane interactions (see above), however this does not impact aE11 binding or the neoepitope.

If so, it would be helpful for the authors to describe these.

The following point has been added to the main text to address this point (as well as that from reviewer #4).

[Line 135]: “Given that aE11 is known to recognise the oligomeric C9 component of MAC, we chose to employ the polyC9 model. PolyC9 is also recognised by aE11 and its structure closely resembles MAC (21, 22, 31).”

21. Menny A, Serna M, Boyd CM, Gardner S, Joseph AP, Morgan BP, et al. CryoEM reveals how the complement membrane attack complex ruptures lipid bilayers. *Nat Commun.* 2018;9(1):5316.

22. Spicer BA, Law RHP, Caradoc-Davies TT, Ekkel SM, Bayly-Jones C, Pang SS, et al. The first transmembrane region of complement component-9 acts as a brake on its self-assembly. *Nat Commun.* 2018;9(1):3266.

31. Dudkina NV, Spicer BA, Reboul CF, Conroy PJ, Lukoyanova N, Elmlund H, et al. Structure of the poly-C9 component of the complement membrane attack complex. *Nat Commun.* 2016;7:10588.

2) Figure 2 is very crowded and would benefit from additional spacing between cartoons.

We have increased the spacing around panels in figure 2 and improved labels for better clarity.

3) It would be helpful for the general audience to give a sentence or two explanation on how to read/interpret the C9 blot and to improve labelling of Figure 4A. It is not immediately clear that this is a slot blot and therefore difficult to interpret at first. For example, reference to observations at the different concentrations applied would be helpful.

We have added a label in panel (a) and modified the figure legend to avoid confusion with Western blots.

The following point is made in the figure legend to guide readers:

“A concentration series of human, murine and disulphide trapped C9 variants in the monomeric state detected by aE11 IgG in a slot immunoblot assay. An arrow shows the approximate physiological concentration of monomeric C9 (900±200 nM).”

4) Figure 4D and E – if I understand correctly, the data plotted in E is the max response (150nM) plotted in D. Small suggestion, it would be helpful if colors for each mutant were consistent between D and E as at first sight it appears the data do not correspond.

This error has been fixed.

5) The discussion would benefit from citing a few recent studies that have used aE11 to detect MAC in functional experiments (e.g. detection of MAC in AChR loss *in vitro* / bacteria / macrophages or endothelial cells). This would add weight to the importance and impact of the work presented.

As suggested by the reviewer, we have elaborated on the opening paragraph of the discussion and included the relevant studies to emphasise the importance of aE11:

[Line 355]: “The MAC is a key immune effector in pathogen elimination and has been associated with a variety of different inflammatory and autoimmune diseases. As such, quantifying the level of pores *in vivo* and *in vitro* provides insight into the contributions of the MAC to these diseases. The monoclonal antibody, aE11, recognises a neoepitope in the oligomeric C9 component of the MAC, and therefore acts as a specific marker for the quantification of MAC (simplified schematic; **Fig 5**). Historically, aE11 has been used to detect C5b-9 deposition in a number of diseases, both experimentally and clinically. For example, aE11 has been used to detect MAC deposition associated with cancer, bowel, neurological and kidney diseases (34-40), as well as MAC in macrophage inflammasome activation (41). Finally, in the development of the total complement activity ELISA, aE11 was compared with several other anti-C9 neoepitope antibodies and found to be the most specific and the one with highest affinity (42).”

34. Halstensen TS, Mollnes TE, Brandtzaeg P. Persistent complement activation in submucosal blood vessels of active inflammatory bowel disease: immunohistochemical evidence. *Gastroenterology*. 1989;97(1):10-9.

35. Halstensen TS, Mollnes TE, Fausa O, Brandtzaeg P. Deposits of terminal complement complex (TCC) in muscularis mucosae and submucosal vessels in ulcerative colitis and Crohn's disease of the colon. *Gut*. 1989;30(3):361-6.

36. Halstensen TS, Mollnes TE, Garred P, Fausa O, Brandtzaeg P. Epithelial deposition of immunoglobulin G1 and activated complement (C3b and terminal complement complex) in ulcerative colitis. *Gastroenterology*. 1990;98(5 Pt 1):1264-71.
37. Halstensen TS, Mollnes TE, Garred P, Fausa O, Brandtzaeg P. Surface epithelium related activation of complement differs in Crohn's disease and ulcerative colitis. *Gut*. 1992;33(7):902-8.
38. Koopman JJE, van Essen MF, Rennke HG, de Vries APJ, van Kooten C. Deposition of the Membrane Attack Complex in Healthy and Diseased Human Kidneys. *Front Immunol*. 2020;11:599974.
39. Kolka CM, Webster J, Lepletier A, Winterford C, Brown I, Richards RS, et al. C5b-9 Membrane Attack Complex Formation and Extracellular Vesicle Shedding in Barrett's Esophagus and Esophageal Adenocarcinoma. *Front Immunol*. 2022;13:842023.
40. Mollnes TE, Vandvik B, Lea T, Vartdal F. Intrathecal complement activation in neurological diseases evaluated by analysis of the terminal complement complex. *J Neurol Sci*. 1987;78(1):17-28.
41. Diaz-Del-Olmo I, Worboys J, Martin-Sanchez F, Gritsenko A, Ambrose AR, Tannahill GM, et al. Internalization of the Membrane Attack Complex Triggers NLRP3 Inflammasome Activation and IL-1beta Secretion in Human Macrophages. *Front Immunol*. 2021;12:720655.
42. Seelen MA, Roos A, Wieslander J, Mollnes TE, Sjöholm AG, Wurzner R, et al. Functional analysis of the classical, alternative, and MBL pathways of the complement system: standardization and validation of a simple ELISA. *J Immunol Methods*. 2005;296(1-2):187-98.

Reviewer #4 (Remarks to the Author):

The manuscript entitled “The neoepitope of the complement C5b-9 Membrane Attack Complex is formed by proximity of adjacent ancillary regions of C9”, by Bayly-Jones et al., describes the interaction between the poly-C9 structure, as a mimic of the complement MAC complex, and the monoclonal antibody aE11. The 3.2 Å resolution cryoEM structure of the complex between the complement protein C9 and the aE11 Fab is shown and used to refine a molecular model that allow the authors to identify the structural determinants of the aE11-C9 interaction and shed light in the antibody recognition mechanism. Furthermore, the interaction is thoroughly studied through biophysical methods and the use of several C9 mutant variants. Specificity of the aE11 antibody for the oligomerized C9 versus the monomeric stage is also discussed.

General comments

The manuscript shows a detailed analysis of the polyC9-aE11 complex, although it does not emphasize the scope of their conclusions. A more extensive discussion about the application of the recognition mechanism discovered could include some ideas such as engineered therapeutic antibodies to inhibit the polyC9 (MAC) formation or antibodies with enhanced affinity and/or specificity for C9 oligomers, etc.

Although the data was unpublished, initial characterisation of aE11 revealed that it does not have an inhibitory effect on lysis. This, along with strategies to produce inhibitory antibodies against the MAC, was elaborated in the discussion:

[Line 400]: “As we have discovered that the antibody binds to the periphery of the pore, we expect that this antibody alone does not contribute to obstructing pore formation. Indeed, the

initial characterization of aE11 revealed that it did not inhibit lysis (unpublished data). Furthermore, to the best of our knowledge, no other anti-C9 neoepitopes have been shown to inhibit lytic activity. Foundations for producing a potentially inhibitory antibody may involve binding to a major interface (e.g., the MACPF/CDC) to block oligomerisation, or capturing a conformational change. As of recent, it seems that inhibition of C7 is the last step in the terminal pathway that can inhibit lysis efficiently (45).”

45. Zelek WM, Morgan BP. Monoclonal Antibodies Capable of Inhibiting Complement Downstream of C5 in Multiple Species. *Front Immunol.* 2020;11:612402.

Further into the discussion, we have mentioned that our findings serve as the structural basis for modifying aE11 to recognise C9 oligomers from different species to study terminal complement-driven diseases, such as mice models.

[Lines 417-419]: “Variants of aE11 that recognise murine MAC would be beneficial and would serve to expand the repertoire of tools to study MAC-driven inflammatory disease.”

Due to the limited applicability of this monoclonal antibody (as explained along the manuscript), the rationale for the selection of this specific antibody in this work remains unclear to me. Are there any other monoclonal C9 antibodies available with other relevant properties?

Although there are other antibodies that also recognise the MAC (outlined in reviews by Koopman *et al.*, 2022 and Harboe *et al.*, 2011), aE11 is the most relevant as it is (1) found to be the most specific and stable against MAC and (2) it has worldwide applications in a complement ELISA to screen for complement deficiencies/overactivation. This was discussed in response to reviewer #3 comment 5.

From a practical standpoint, we have access to large quantities of purified aE11 material through our collaboration with the original authors who discovered aE11 (authors C.L. and T.E.M.). Obtaining these quantities of other antibodies via commercial sources would be prohibitively expensive.

It would be helpful if the authors could discuss in more detail to what extent the present work is applicable to the physiological MAC complex. The major use of the aE11 antibody is the specific recognition of the MAC in plasma as an indicator of inflammatory and auto-immune diseases. However, the analysis is performed only with the polyC9 complex and no experiments refer to the native MAC or sC5b-9.

Please see response to reviewer #3 regarding the relevance of polyC9 with respect to MAC.

We have included a new paragraph in the methods section to describe the MAC formation assays we performed (Fig 2f), as well as improving labelling of Fig 2f and figure legend to prevent misunderstanding.

“f. Slot immunoblot of aE11 binding to the oligomeric human C9, but not oligomeric mouse C9, component of whole MAC. MAC assembly intermediates (C5b6, C5b-7, C5b-8) are included as controls. Concentrations of C5b6 are marked for each condition. The C7, C8 $\alpha\beta\gamma$, and C9 concentrations were constant at 30 nM, 30 nM, and 7 nM respectively.”

Specific comments

1. Line 121. I think that the sentence “The aim of this study is ... investigate the necessary conformational changes that occurs during pore assembly” is confusing. The analysis of this conformational changes is focused on the aE11 interaction surface and not on the conformational changes undergone by C9 during the pore assembly.

We have rephrased:

“The aim of this study was to assess the structural basis for the formation of the C9 neoepitope and determine whether conformational changes due to pore formation are required.”

2. Line 127. Although the title is “Structural characterization of the C9 neoepitope in MAC by cryo-EM”, no MAC structures are shown. I think the title must refer to the actual specimen (poly-C9) instead.

We have modified the title:

“Structural characterisation of the C9 neoepitope in oligomeric C9 by cryo-EM.”

3. The initial consensus refinements (C1 and C22 reconstructions of the complete poly-C9-aE11 complex) mentioned in the text could be included in the EMDB deposition as accessory maps. That would help to understand statements of the manuscript such as the “stoichiometric binding of aE11-Fab” (line 134) or the “intrinsic heterogeneity of polyC9” (line 137).

We have deposited these consensus reconstructions as requested. We have also included additional references to supplementary figure 1a where the heterogeneous nature of the specimen is clearly observable in the micrograph.

4. Figure 1a. Showing complete 2D average images (unmodified boxes used along the 2D alignment and classification) instead of cropped images, as well as the corresponding scale bars, would help to compare both structures. Diffuse densities described in lines 135-137 might be indicated in the corresponding panel to help the structure interpretation.

Scale bars have been added in the main figure. The requested complete set of 2D class averages are available in the corresponding supplementary figure (supp fig 1b), these highlight the diffuse signal described.

The most meaningful comparison is by 3D volumes, which is also shown in panel (a), since 2D projections can be subjective and difficult to interpret due to the incompleteness problem. As stated above, the consensus maps have also been deposited allowing the reader to inspect the variability in the density maps themselves.

5. Line 165. I would recommend to start with the description of the 2D average images in the figure legend to help figure interpretation.

We have modified the figure legend as suggested for clarity:

“[...] a. Above: comparisons of 2D class averages between polyC9 alone and aE11-Fab/polyC9. Below: 3D density map comparison between top-down and side [...]”

6. Figure 2b legend does not help to follow the panel order and description. There is no relative movement information regarding the aE11-polyC9 atomic models. Some of the colors used in panels c, d and e are not easy to distinguish. To highlight the residues selected for the C9 point mutations would ease their identification.

The figure legend has been re-written to improve the readability.

“Key regions of antibody binding to C9. Locations of these regions correspond to the boxed regions in (b). Mutated residues for validation studies are encircled in a red outline. f. Slot immunoblot of aE11 binding to the oligomeric C9 component of whole MAC. Oligomeric human C9 is recognised by aE11, but not oligomeric mouse C9. MAC assembly intermediates (C5b6, C5b-7, C5b-8) are included as controls. Concentrations of C5b6 are marked for each condition. The C7, C8 $\alpha\beta\gamma$, and C9 concentrations were constant at 30 nM, 30 nM, and 7 nM respectively. g. Sequence alignment of the linear regions of human and murine C9 that contribute to the quaternary discontinuous epitope.”

Relative movement of atomic models is shown in supplementary figure 2 - however we observed virtually no changes and thus these comparisons are largely uninformative.

We have highlighted the mutated residues in panels c, d, and e with a red circle.

7. Line 227. Although the analysis of the thermal stability of the C9 mutants points to no structural alterations in the overall C9 structure, it cannot be ruled out that these mutations affect in some way the structure of the interaction interface. Some extra discussion about the predicted structural alteration due to the nature of the introduced mutations might be interesting.

To definitively show that the epitope is unchanged (with the exception of the point mutation which of course will be necessarily different), we would need to determine structures of all the C9 variants - unfortunately this is beyond the scope of a single study - but we agree it would be interesting.

Fortunately, our functional studies (haemolytic, nanoDSF and NS-EM) suggest that overall the protein is still folded and retains normal function. Ultimately the aim of these mutations is to understand how those residues affect aE11 binding and this also encompasses the effects on the local fold (which indeed may contribute to the reduced binding).

We have added the following discussion points:

[Line 247]: “Murine residues were chosen such that the impact on the local epitope fold would be minimised, assuming that human C9 should tolerate murine C9 substitutions.”

[Line 255]: “On the basis of these analyses alone, we cannot fully disregard the possibility that these mutations cause subtle local changes in the epitope.”

8. Figures 3d and 3e shows a significantly higher SEM for the wildtype than the mutants that might be worth mentioning in the text.

This is most likely due to differences in total polyC9 quantity as batch to batch variation occurs during the oligomerisation reaction. Discussing this variation does not provide any new insight or change our conclusions.

In, figure 3e, the dashed rectangle showing “no binding” might be misleading with respect to the expected response.

There is essentially no concentration-dependent change in the sensorgrams of R65Q/V68E/P72 (visible in Fig 3d), this was interpreted as “no binding”. We have removed the dashed box to avoid confusion and simply shown the mean \pm std. dev. of binding (3.17 ± 1.5 arb. units). We have also included an inset in supplementary figure 6 to more clearly show the raw sensorgram values.

9. Figure 4b does not included the samples color code (neither in the corresponding figure legend).

Figure 4b has now been labelled accordingly.

10. Further comparison between the recombinant monomeric C9 and the native plasma C9 that could be responsible for the differential binding specificities to the aE11 antibody would be interesting. In this regard, comment on the relative concentrations of the recombinant C9 used in these experiments versus the native C9 concentration in plasma would help to understand the discrepancy.

It is reported that aE11 does not recognised monomeric C9 in plasma. We have now conducted further comparisons (included as supplementary figure 8, see below) that reproduces these observations. It appears that high concentrations of serum albumin (and/or other factors) significantly reduce aE11 binding below the detectable limit probably in a non-specific manner.

We found that adding detectable concentrations of purified monomeric C9 to depleted human serum (that does not contain C9) resulted in complete loss of aE11 detection, suggesting that the weak affinity to the monomer is insufficient to drive complex formation in the context of serum.

Results:

[Line 333]: “Previous studies report that aE11 is unable to detect monomeric C9 in serum. Thus to reconcile the unexpected finding of aE11 binding *in vitro*, we conducted a slot-immunoblot assay to compare recombinant and native C9, both *in vitro* and in the context of human serum. As expected, aE11 recognised both recombinant and native C9 with comparable levels (**Supp Fig 8**). Conversely, aE11 did not detect native C9 in serum, nor could it recognise recombinant C9 that had been supplemented back into C9 depleted serum at equivalent concentrations (**Supp Fig 8**). This indicates that the weak binding of aE11 to monomeric C9 is abolished in the context of serum.”

Discussion:

[Line 376]: “Lastly, we unexpectedly observed aE11 is also capable of binding to purified soluble monomeric C9 in a concentration dependent manner *in vitro*, albeit at much lower affinity (summarised in **Fig 5**). This interaction was furthermore found to be specific as site-directed mutagenesis to the C9 neoepitope had predictable and expected impact on the interaction. While we were able to detect this interaction at physiological concentrations ([C9] in plasma $\sim 900 \pm 200$ nM), we note studies which report that plasma C9 is undetectable by aE11 (4, 25, 42). Indeed, we observed a serum-dependent loss of binding when supplementing high concentrations of monomeric C9 back into C9-depleted serum (Supp. Fig 8). This is consistent with observations in the literature (26, 42, 44) however the exact cause of this effect remains to be characterised.”

4. Harboe M, Thorgersen EB, Mollnes TE. Advances in assay of complement function and activation. *Adv Drug Deliv Rev.* 2011;63(12):976-87.
25. Mollnes TE, Lea T, Harboe M, Tschopp J. Monoclonal antibodies recognizing a neoantigen of poly(C9) detect the human terminal complement complex in tissue and plasma. *Scand J Immunol.* 1985;22(2):183-95.
26. Mollnes TE, Lea T, Froland SS, Harboe M. Quantification of the terminal complement complex in human plasma by an enzyme-linked immunosorbent assay based on monoclonal antibodies against a neoantigen of the complex. *Scand J Immunol.* 1985;22(2):197-202.
42. Seelen MA, Roos A, Wieslander J, Mollnes TE, Sjöholm AG, Wurzner R, et al. Functional analysis of the classical, alternative, and MBL pathways of the complement system: standardization and validation of a simple ELISA. *J Immunol Methods.* 2005;296(1-2):187-98.
44. Bergseth G, Ludviksen JK, Kirschfink M, Giclas PC, Nilsson B, Mollnes TE. An international serum standard for application in assays to detect human complement activation products. *Mol Immunol.* 2013;56(3):232-9.

Supp Figure 8. Slot immunoblot of aE11 binding to monomeric purified C9 or human serum. A concentration series of human (Hu) and murine (Mu) C9 variants in the monomeric state, serum and C9-depleted serum that was supplemented with recombinant C9 detected by aE11 IgG in a slot immunoblot assay. Native C9 was diluted into C9 depleted serum to a final concentration of 3170 nM, serial dilution was conducted with C9 depleted serum (Comp Tech). *Since C9 concentrations in serum is unknown, it was initially added undiluted and then serially diluted 2-fold; the approximate physiological concentration of monomeric C9 in neat plasma is $\sim 900 \pm 200$ nM

11. Figure 5. It is not clear what the “Moderate” and “Strongest” panels add to the figure. Some colors (C8 γ and C9) are indistinguishable. Image quality should be improved.

These illustrate the relative binding strength of each interaction. We have included additional labels to make this clearer.

The colour of C8 γ was changed to distinguish it from C9.

12. Line 360-361. Are the C9 in vitro concentrations used to see the aE11-monomeric C9 binding similar to the physiological monomeric C9 concentrations in plasma?

Yes - The physiological monomeric C9 concentration in plasma is roughly 900 nM (Oleesky *et al.*, 1986 and Morgan 2018). In the slot immunoblot assays of monomeric C9, we tested a range below, above, and inclusive of the physiological level (100 nM to 3 μ M). A lower range was tested in SPR experiments (30 to 150 nM). We have added an arrow to illustrate this in figure 4a.

See responses above.

13. In a similar way to the analysis carried out on how the interaction of this antibody does not impair the poly-C9 (MAC) assembly, I think it would worth analyzing whether it might be helping to stabilize the C9 oligomers and, more interestingly, the MAC assembly and

activation.

Comparison of the reconstruction to previous reconstructions (without aE11) suggests that some small loops might be stabilised by aE11 binding (e.g. the TSP-LDLRA linker), since these loops are resolved in the data presented here. However, this may not necessarily reflect stabilisation, rather this may relate to the processing strategy or improved sample quality.

“Notably, the aE11 bound structure possesses improved cryo-EM density for the TSP and LDLRA domains compared to our previous reconstruction enabling this region to be modelled (Fig 1 d).”

14. Line 365. Although the reference to the C9 protein expression and purification is included, I would at least mention the expression system used in the main text, due to its relevance.

We have now included these details.

[Line 429]: “All C9 variants were expressed using the Expi293F mammalian expression system and purified using tag-less purification methods as previously described (47). Likewise, oligomerisation of C9 to polyC9 was induced as previously described (21).”

15. Line 381. Extinction co-efficient units should be included.

We have added units for the extinction coefficients.

[Lines 446-447]: “... the extinction co-efficient [...] was taken as $8.99 M^{-1} cm^{-1}$ ” and “the murine C9 was taken as $8.85 M^{-1} cm^{-1}$ ”.

16. Line 382. “Murine” should be in lowercase.

Fixed.

17. Line 444. Radiation damage cannot be corrected with MotionCor2.

Radiation damage due to inelastic scattering is manifest as a dose-dependent decay of high-resolution spatial features, this can be (partially) corrected for by per-frame dose-dependent frequency weighting (or simply dose weighting). However, we agree that technically speaking MotionCor2 cannot perfectly “correct for” beam induced motion nor radiation damage. Ultimately we can only partially compensate for both of these effects. We have rephrased the methods, however this is mostly a semantic difference.

[Line 512-513]: “Beam induced motion was corrected by MotionCor2 with dose-dependent weighting to compensate for radiation damage (39).”

18. Line 446. Although the pixel size is shown in one Supp Table, images box size expressed in Å help interpretation.

Box sizes in pixels are most relevant to image analysis as this is the conventional unit used during data processing. For the benefit of the reader scale bars have been included in figures for the interpretation of dimension.

19. Supp Figure 1d shows a significant preferential orientation. Albeit there are frontal views in the consensus refinement (as shown in the average images in panel b), they do not seem to contribute in the density subtracted reconstruction. Comment on this would help to understand the refinement procedure.

In fact, all views are included in the reconstruction (the class averages are not necessarily equally populated) and when these are smeared across the full angular space, they become less visible.

This preferential orientation would normally be challenging to overcome for a C1 particle, however since this reconstruction is derived from a consensus reconstruction (with C22 symmetry imposed) the strong bias appears to have little effect on the isotropy of the final reconstruction.

The cryoSPARC angular distribution below illustrates the additional views differently.

20. Supp Figure 3 displays an outline. In Supp Fig 3b, color code for the coulombic surface is omitted. Charge complementary discussed would be clearer if the complementary areas are indicated as they have been included in panel c.

We have modified the figure as suggested.

21. Line 566. “versus” should be in italics.

Fixed.

22. Supp Figure 6. Colors of the lines in the legend are not easy to distinguish. Thicker lines might help to clarify.

We have modified the figure to improve the visibility.

23. Table caption in Supplementary material would help to understand abbreviations used in them.

We have now included Supplemental table 4 captions with appropriate abbreviations.

Other minor changes:

[Methods]

We have noticed that we have not included one method description and have added it for clarity.

Formation of MAC on liposomes

Liposomes of *E. coli* total lipid extract (ETL, Avanti Polar Lipids) were generated as follows. Chloroform-solubilised lipid was dried under Argon gas in a clean glass tube and then desiccated under vacuum overnight at room temperature. Subsequently, the lipid film was rehydrated and resuspended in HBS (10 mM HEPES-NaOH, pH 7.2, 150 mM NaCl) to a final concentration of 0.5 mg mL⁻¹ and sonicated until large chunks were no longer visible. Finally, the lipid mixture was extruded via a polycarbonate membrane with pore size of 0.4 µm (Avanti Polar Lipid) to form unilamellar liposomes. To form MAC on liposomes, a total of 200 µL of liposome mixture (diluted 1:200 in HBS) was incubated sequentially with each MAC component followed by a 5 min incubation at 37 °C. Initially, liposomes were incubated with a C5b6 (at various final concentrations from 1-16.8 nM), followed by C7, C8, and lastly C9 added to a final concentration of 30, 30 and 7 nM respectively. C5b6, C7, C8 were purchased (Complement Tech) and C9 was purified as described above. After each incubation a 10 µL sample was taken for slot immunoblot.

REVIEWERS' COMMENTS:

Reviewer #3 (Remarks to the Author):

The authors have addressed all my comments and with revisions requested by other reviewers made significant improvements to the manuscript.

Reviewer #4 (Remarks to the Author):

The new version of the manuscript entitle "The neoepitope of the complement C5b-9 Membrane Attack Complex is formed by proximity of adjacent ancillary regions of C9" answered most of the concerns mentioned in my review report. Authors have adequately replied to my comments and improvements done both in the main text and in the figures clarify the results and add some relevance to the paper scope and research applications.